# Research

behaviour, cognition, ecology

New Caledonian crows, planning, spoon test, comparative cognition, future reward

**Author for correspondence:**
M. Boeckle
e-mail: markus.boeckle@gmail.com

†These authors contributed equally to this study.

# New Caledonian crows plan for specific future tool use

M. Boeckle[1,2,3], M. Schiestl[4,5,7], A. Frohnwieser[1], R. Gruber[4], R. Miller[1], T. Suddendorf[6], R. D. Gray[4,5], A. H. Taylor[4,†] and N. S. Clayton[1,†]

[1]Department of Psychology, University of Cambridge, Cambridge, UK
[2]Karl Landsteiner University of Health Sciences, Krems an der Donau, Austria
[3]Department of Psychiatry and Psychotherapy, University Hospital Tulln, Tulln, Austria
[4]School of Psychology, University of Auckland, Auckland, New Zealand
[5]Max Planck Institute for Evolutionary Anthropology, Leipzig, Germany
[6]School of Psychology, University of Queensland, Brisbane, Australia
[7]University of Veterinary and Pharmaceutical Sciences, Brno, Czech Republic

  MB, 0000-0002-0738-2764; AF, 0000-0001-5219-4319; RG, 0000-0002-7313-7396; RM, 0000-0003-2996-9571; TS, 0000-0003-3328-7442; RDG, 0000-0002-9858-0191; AHT, 0000-0003-3492-7667; NSC, 0000-0003-1835-423X

The ability to plan for future events is one of the defining features of human intelligence. Whether non-human animals can plan for specific future situations remains contentious: despite a sustained research effort over the last two decades, there is still no consensus on this question. Here, we show that New Caledonian crows can use tools to plan for specific future events. Crows learned a temporal sequence where they were (a) shown a baited apparatus, (b) 5 min later given a choice of five objects and (c) 10 min later given access to the apparatus. At test, these crows were presented with one of two tool–apparatus combinations. For each combination, the crows chose the right tool for the right future task, while ignoring previously useful tools and a low-value food item. This study establishes that planning for specific future tool use can evolve via convergent evolution, given that corvids and humans shared a common ancestor over 300 million years ago, and offers a route to mapping the planning capacities of animals.

## 1. Background

Can non-human animals plan for specific future situations? Despite a sustained research effort over the last two decades, there is still no consensus on this question [1–15]. The ability to plan for future events is one of the defining features of human intelligence [1,16,17]. The extent to which this ability is unique to our species has been hotly debated for over two decades [1–9,18]. The main reason for this is that alternative explanations can account for the reported animal successes. Consider the most prominent task, 'the spoon test' [17,19]: to pass the spoon test, the subject must select a tool for an event that might happen in the future. Typically, there is a single choice from a number of objects, of which only one can be used to solve the problem. Both apes and corvids have been shown to ignore distractor objects and instead choose the functional object, thereby passing the test [2,6,8]. However, there have been concerns that choices could be driven by the value of the target object in the present being higher than those of the distractor objects, rather than by the animal imagining the future utility of the tool [1,7,9,20–22]. These concerns have persisted despite some attempts to address them or rule them out (e.g. [6,8,23,24]), and recently received empirical support from a study on children [22]. After seeing a specific problem, the children were presented with two objects that had high value. One of these objects could be used to solve the observed problem, while the other could not. Children under the age of five chose at chance between these

objects, despite clearly being able to remember which problem they had observed. These results demonstrate that associative learning can drive successful performance on the spoon test, rather than the use of foresight, thereby substantiating the possibility that previous spoon test studies may have reported false positives for the presence of planning in animals.

One way to provide more compelling evidence of planning would be to present animals with a more stringent test where they have to choose between multiple tools after observing a specific problem being set up, such that the same objects function as solution in one condition and as distractors in another [7]. In this situation, each tool would have high value due to it being associated with positive outcomes in the past, but would only be useful when the correct problem was available in the future. By varying the problem that will be available in the future, it would be possible to see if an animal can choose the correct tool for a particular anticipated task. Successful solution of such tasks would demonstrate that an animal is capable of planning for specific future tool problems. Our study brought together researchers who have published contrasting 'rich' and 'lean' interpretations of animal planning studies to run a pre-registered version of such a test.

## 2. Methods

The methodology has been described previously [25] and was used in a similar way to test flexible planning in young children [26].

### (a) Subjects

New Caledonian crows were housed for 5 months in an outside aviary on the island of Grand Terre, New Caledonia. Based on the sexual size dimorphism [27] four of the nine crows were females (Mercury, Neptune, Triton, Uranus) and five were male (Io, Jupiter, Mars, Saturn, Venus). Based on the coloration of their beaks, five of the crows were juveniles under one year of age (Neptune, Triton, Jupiter, Mars, Venus) and four were adults over 2 years of age (Mercury, Io, Saturn, Uranus). The aviary had 10 cages, each measuring at least $2 \times 3 \times 3$ m. Access to water was granted ad libitum. The general diet was fruits and soaked dog food. Pieces of meat functioned as rewards during training and testing. All testing took place in two compartments that were visually inaccessible to each other and the other crows. Our work was carried out under the approval of the University of Auckland Animal Ethics Committee (reference no. 001823).

#### (i) Participation of individuals across trials

Nine individuals entered training. Three individuals (Jupiter, Io, Mercury) did not reach criterion in the tool functionality training and were therefore excluded from later procedures and testing. Two crows were unable to make correct choices in Conditions 1 and 2 (Mars, Venus). The remaining subjects took on average 22 trials to reach training criterion. Four individuals (Saturn, Neptune, Triton, Uranus), reached training criterion at C1 and C2 and then entered the testing phase (C3 and C4) and three individuals were tested in the follow-up (Neptune, Triton, Uranus).

### (b) Apparatus

We used three types of apparatus, namely a remote-controlled feeder apparatus (the dispenser apparatus), a stone dropping collapsible platform box (the platform apparatus) and a horizontal Perspex tube (the tube apparatus). The dispenser apparatus consisted of a wooden box $33 \times 30 \times 20$ cm with a $6.3 \times 3$ cm

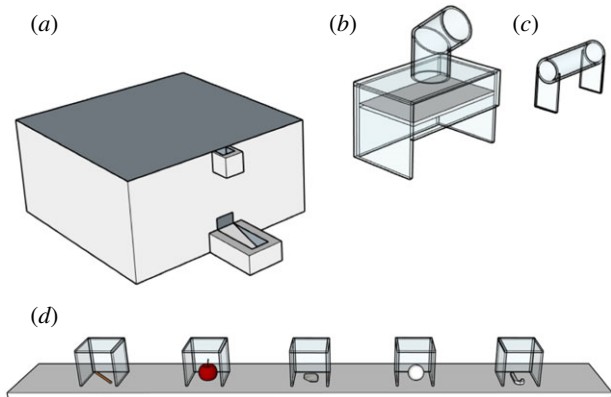

**Figure 1.** The three apparatuses used in the study: (a) dispenser apparatus, (b) platform apparatus, (c) tube apparatus. (d) The tool presentation box, including a stick, stone and hook as tools, apple as a lower-quality food reward and a ball as a distractor object. The ball as a distractor was introduced one day before the first presentation of the five-choice tool functionality training. (Online version in colour.)

slot in its top surface, into which the crow could insert items (figure 1). It contained a disc, which turned when activated by a remote-control button, dispensing one piece of food. Birds were trained to drop a hook tool into the object slot to get food. The platform apparatus was a $16 \times 10 \times 10$ cm transparent Perspex box of the same design as that used in past work on physical cognition in corvids [28–31]. It had a collapsible trap-platform within the box that released a food reward when a stone was dropped onto it. To prevent stick tools from being able to release this mechanism, we installed a 12 cm long tube with a diameter of 5 cm and a slant of 30° in the middle. Birds were trained to drop stones into this apparatus. The tube apparatus was a horizontal Perspex tube, 18 cm long, with a diameter of 5 cm, mounted 8 cm above a base. Birds had to insert a wooden stick tool to push or pull the meat reward out of the tube.

### (c) Procedure

All birds participated in various experiments before the presented study [31–33]. The training specifically required for the current study are tool use training, tool selection training, apparatus functionality training, hook training and tool transport training, five choice tool functionality training, and the temporal sequence training, which is outlined in the description of Conditions 1 and 2. For a detailed and complete description of the prior experience and specific training stages, see below and electronic supplementary material.

#### (i) Training phase

New Caledonian crows were first trained to use three tool-apparatus combinations (stick to tube, stone to platform, hook to dispenser; figure 1). We used two compartments; one in which the baited apparatus was presented without the tool and one where the choice between objects was presented (figure 2). Compartments were directly adjacent to each other but did not allow visual access to each other when the connecting door was closed.

Then birds were trained that a specific temporal sequence would occur during the experiment (Conditions 1 and 2). Conditions 1 and 2 were run with the tube apparatus. In Stage 1 they would be shown a baited apparatus for 1 min. After this, the bird was moved to the next-door compartment and the connecting door was closed so the crows had no visual access to the other compartment. After 5 min in this compartment, the birds were presented with the tool presentation box containing five

Proc. R. Soc. B 287: 20201490

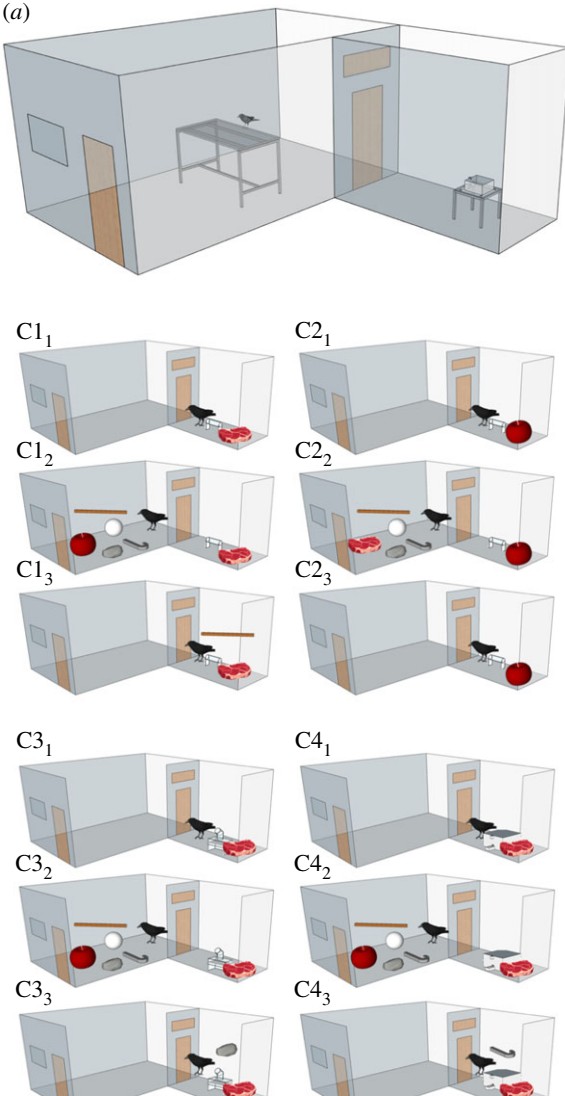

**Figure 2.** Compartment set-up. (*a*) The test compartments. The left compartment contains a table, on which the tool presentation box was placed in the choice phase. The right compartment contains a small table where the apparatus was placed at inspection phase and accession phase. Crows could move between compartments when a sliding door was opened. Condition 1: crows observe a tube baited with meat ($C1_1$) for 1 min, after which they are moved to the left compartment. After 5 min, they are presented with the three tools, a distractor and low value apple ($C1_2$). Once a choice has been made the tool presentation box is removed. After 10 min the door to the right compartment is opened, allowing the crows to access the food if they have chosen the stick ($C1_3$). Condition 2: crows observe that the tube is baited with low value apple ($C2_1$), and then are presented with the choice of three tools, a distractor and meat ($C2_2$). Once a choice is made the presentation box is removed, and after 10 min the crows are allowed access to the apparatus ($C2_3$). Conditions 3 and 4: test conditions. Crows are given alternating trials of $C3_{1-3}$ and $C4_{1-3}$, where they see the platform apparatus ($C3_1$) or the dispenser apparatus ($C4_1$) baited with food, and then are moved to the next door compartment, where, 5 min later, they are presented with the tool presentation box containing three tools, a distractor and low value apple ($C3_2$ and $C4_2$). To gain the meat the crows need to choose a stone in $C3_2$ and the hook in $C4_2$, so they can take this tool to the apparatus in the next door compartment 10 min later ($C3_3$ and $C4_3$). We trained the birds in C1 and C2 to understand that the specific future event will differ from the next one and that they have to be attentive to the presented apparatus. This is one of the critiques of the study by Kabadayi & Osvath [8]. Thus, the actual test is when birds have 'learnt' the temporal rule and then in the test phase are presented with new apparatus–tool combinations they have to plan for. (Online version in colour.)

objects: a stick, a hook template, a stone, a distractor object and a very small piece of apple (Stage 2). Positions of the objects with the box were pseudorandomised across trials. After making a choice, this apparatus was removed. Ten minutes later the door was re-opened to the next-door compartment (Stage 3) and given access to the baited apparatus they had observed in Stage 1. Thus, birds were allowed to take the tool they had chosen in Stage 2 to the apparatus.

To train the crows on this sequence of events, in Condition 1, we placed highly valued meat in a long horizontal tube at Stage 1, gave crows the choice of a stick, a hook, a rock, a distractor object and lowly valued apple at Stage 2, and then gave crows access to the meat-baited tube at Stage 3 (figure 2). The optimal choice in Condition 1 at Stage 2 was to take the stick while ignoring the other objects, so it could be used 10 min later to get the meat from the tube in Stage 3. In Condition 2, in contrast, we placed low-value apple in the tube in Stage 1. Crows were then given the choice of a stick, a hook, a rock, a distractor object and highly valued meat at Stage 2, before being presented with the apple-baited tube in Stage 3. The optimal choice in this condition was to ignore the tools and take the meat during Stage 2, as the crows could only get low-value apple in the future if they chose the stick. Trials in which birds who chose meat in Stage 2 were ended after the choice. Four of the six crows tested were able to make correct choices in Conditions 1 and 2, with subjects taking on average 22 trials to learn this. Birds received alternating trials of Conditions 1 and 2 until they reached a criterion of 7/10 correct trials for each of these conditions.

If the birds dropped or placed the tool within the aviary, birds had to retrieve the tool from this location unless it was out of reach. If the tool fell in a position the birds were not able to retrieve, the tool was retrieved by the experimenter and placed at the closest location to the place it fell that was again reachable for the bird, which happened on 10 out of 167 trials (also see table 3). In trials where birds did not choose a tool, the trial was terminated and treated as a wrong choice. In any cases where a wrong tool was chosen, the trial continued and the crows were given the opportunity to interact with the tool and the apparatus.

### (ii) Testing phase

The critical part of our study was the test Conditions 3 and 4, in which the crows were presented with trials involving the same temporal sequence. Testing conditions were identical to the training phase except for the identity of the apparatus that was presented at Stage 1. In training, this had always been the tube apparatus.

In Condition 3, we set up a drop-down platform apparatus operated by a stone at Stage 1, meaning crows now had to choose the stone in Stage 2 while ignoring the other objects and the low-value apple in order to obtain the meat at Stage 3. In Condition 4, we set up a dispensing apparatus operated by a hook, meaning crows now had to choose the hook while ignoring the other objects and the low-value apple. We chose apple as low-value immediate reward based on a preference test conducted in a previous study, in which they chose apple over tools [34]. Crucially then, to get their preferred food item, the crows had to select a tool appropriate for the apparatus they had reason to expect, through their experience in the training conditions, would be available in the future. They had never before experienced these test conditions, and had to ignore the object that now had the highest value, the stick, which had been associated more with food in Conditions 1 and 2 than the other choices. The crows could only take advantage of the future event that would occur 10 min later in Stage 3 if they chose, at Stage 2, the specific tool required for the apparatus that they had seen set up in Stage 1, and which critically, they had had no further visual access to. Each of the four crows that passed Conditions 1 and 2 were given alternating trials of Condition 3 and Condition 4 until they had received

**Table 1.** Performance of individuals. Mars and Venus did not pass criterion in Conditions 1 and 2.

| | individual | correct | total | % | CI— | CI+ | effect size | binomial *p* |
|---|---|---|---|---|---|---|---|---|
| conditions 1 and 2 | Mars | 22 | 40 | 55 | 0.38 | 0.70 | 0.55 | ≤0.001* |
| | Venus | 16 | 30 | 53 | 0.34 | 0.72 | 0.54 | ≤0.001* |
| | Neptune | 17 | 20 | 85 | 0.62 | 0.97 | 1.33 | ≤0.001* |
| | Saturn | 19 | 23 | 83 | 0.61 | 0.95 | 1.16 | ≤0.001* |
| | Triton | 18 | 21 | 86 | 0.64 | 0.97 | 1.41 | ≤0.001* |
| | Uranus | 19 | 25 | 76 | 0.55 | 0.91 | 0.83 | ≤0.001* |
| conditions 3 and 4 | Neptune | 9 | 10 | 90 | 0.55 | 0.99 | 2.00 | ≤0.001* |
| | Saturn | 3 | 10 | 30 | 0.07 | 0.65 | 0.29 | 0.429 |
| | Triton | 7 | 10 | 70 | 0.35 | 0.93 | 0.67 | ≤0.001* |
| | Uranus | 7 | 10 | 70 | 0.35 | 0.93 | 0.67 | ≤0.001* |

**Table 2.** Results of full generalized linear mixed model looking at learning effect in training and testing. When individuals that did not pass criterion in Conditions 1 and 2 are excluded ($n = 4$) a significant training effect can be shown. With the two individuals ($n = 6$), no learning effect is present as the effect trial is not significant and the base model without trial as fixed factor is not different from the model with trial included ($p \geq 0.05$). St.E, standard error.

| | | estimate | St.E | CI— | CI+ | z | p |
|---|---|---|---|---|---|---|---|
| training $n = 4$ | intercept | 0.405 | 0.537 | −0.646 | 1.457 | 0.756 | 0.449 |
| | trial | 0.208 | 0.097 | 0.019 | 0.398 | 2.155 | 0.031 |
| training $n = 6$ | intercept | 0.405 | 0.537 | −0.646 | 1.457 | 2.426 | 0.015 |
| | trial | 0.208 | 0.097 | 0.019 | 0.398 | −0.003 | 0.998 |
| testing $n = 4$ | intercept | 1.925 | 1.033 | −0.100 | 3.950 | 1.863 | 0.062 |
| | trial | −0.392 | 0.268 | −0.917 | 0.133 | −1.465 | 0.143 |

five trials of each. To solve the task and get the food crows had remember what apparatus they had seen 5 min ago during Stage 1 and then select the correct tool during Stage 2, while ignoring the other functional tools, the distractor item and the low value apple. It is important to note that in Conditions 3 and 4 crows observed either a stone or hook apparatus, then 5 min later were given a choice of five objects, and then 10 min later were given access to the apparatus. Crows had never experienced this sequence of temporal events with these objects. When choosing objects, the crows only had the memory of what they had seen at the observation Stage 1 to guide them. If crows were using the relative value of each object, as predicted by an associative learning account, in Conditions 3 and 4 crows should have chosen the object most associated with past reward, namely the stick tool. If subjects were choosing at chance we predicted they would choose the correct object only 20% of the time, given there were five objects to choose from. For a sample video, see [35].

### (d) Statistics

With a 0.2 chance level (choosing one correct object out of the five offered) a bird needed to get five trials out of 10 correct to be above chance at $p < 0.05$ (one-tailed $p = 0.026$) at test or six trials out of 10 correct to be above chance at $p < 0.01$ (one-tailed $p = 0.005$). All statistical tests were conducted in R [36].

## 3. Results

Four of the six crows tested showed performance above chance and reached the criterion in Conditions 1 and 2, taking between 20 and 25 training trials to learn the temporal rule. These four individuals were then tested in the critical test Conditions 3 and 4, where novel tool-apparatus combinations were presented (table 1). Three out of the four tested crows performed significantly above chance across these Conditions 3 and 4, with one subject scoring 9/10 and two subjects scoring 7/10 (binomial choice between five choices, $p < 0.001$). Learning effects in training can be shown when individuals Mars and Venus, which did not pass criterion in Conditions 1 and 2, are excluded (no learning effect when the two individuals are included; table 2). No learning effect can be shown in Conditions 3 and 4. Detailed performances of individuals are presented in table 3.

## 4. Discussion

These results provide evidence that New Caledonian crows can plan for specific future tool use. Across Conditions 3 and 4, three of the four crows tested changed their object choices depending on which apparatus they observed being set up in Stage 1. Therefore, their performance was clearly not based on a preference for a specific tool type but on their observation of which problem they would have available to them in the future.

Our study also shows that New Caledonian crows can learn to use *novel* tools to prepare for specific anticipated events. That is, the crows' choices were made regarding tool behaviours learnt in our aviary rather than tool use behaviours that are

**Table 3.** Choices of individuals per condition and trial. h, hook; s, stick; o, stone; m, meat; a, apple; r, bird did not choose a tool, trial was finished thereafter; ', tool lost and replaced by experimenter; correct choices are highlighted in green, incorrect choices are highlighted in orange.

in this species' repertoire of natural behaviours. Stone tool use has not been observed in any wild New Caledonian crow population, while hook tool use has been observed in other populations [37], but not in the one these crows were sampled from. Hook tool use in our study was also different from wild New Caledonian crow hook use, as in our study it only involved inserting a hook shaped stick into an automated feeding machine, rather than using the hook functionally.

A potential limitation of the study was the low number of individuals that were tested. Owing to the space and time restrictions of the field season, it was not possible to include more than nine individuals in this study, and only four of nine individuals passed the initial training criterion. Interestingly, all individuals that succeeded in the test Conditions 3 and 4 were female, of which, only the individual Uranus was categorized as an adult.

These birds selected the correct tool even though the distractors had been solutions in other conditions or items that could have been more immediately rewarding (i.e. a piece of apple or a ball). We note that the latter were not selected by the birds in Conditions 1–6, even though they had interacted with the ball during a familiarization phase and apple was a daily diet food item, raising the concern that they may have learned to ignore these items over the training trials. However, even when conservatively reanalyzing the results of Conditions 3 and 4 as if the crows had only been offered three options rather than five (and so changing the probability of choosing the correct object by chance from 20 to 33%), we obtain the same finding: three of the four crows performed significantly above chance. Thus, our results are robust to the possibility that the distractor objects did not work as intended. Still, future studies might want to use two different types of low-value food items; one for training and a different one for testing or run a control where the distractor is the optimal choice.

Finally, we cannot completely rule out that crows chose the correct tool because of some kind of associative learning. We do not think this possibility is likely, however, because the birds were trained in C1 and C2 on a different apparatus combination than used during testing in C3 and C4, and we used temporal gaps in our study: the tools were presented 5 min after the presentation of the apparatus (when it was now out of sight) and crows were then only able to gain reward (if they had chosen the correct tool) 10 min after this. Furthermore, tools acted as the functional choice in one trial but as distractor object in the next trial, so the birds could not succeed by simply selecting whatever tool was most recently associated with reward. To strengthen the case further, future studies could run control conditions where the apparatus is visibly removed or destroyed after Stage 1, to examine if the birds would continue to pick the now no longer functional tool, or indicate their understanding by switching to the lower-value apple option.

The crows in our study not only picked a tool that has previously turned out to be useful, but a tool that would be useful for a specific future event. Therefore, New Caledonian crows are a prime candidate for testing the conservative criteria for mental time travel developed by Suddendorf & Corballis [38], which children pass [39]. In addition to testing New Caledonian crows to this standard, there is clearly far more research to be done to understand precisely which cognitive mechanisms underpin the crows' behaviour, and what the extent and limits of nonhuman planning capacities are [40,41].

# 5. Conclusion

Given that corvids and humans shared a common ancestor before 300 Ma our study suggests that planning for specific future tool use can evolve via convergent evolution [42]. Our results also establish a novel study design that could be used effectively to test for future planning in other non-human animals as there is clear evidence that animals can pass this test. Further, we have strived to reach some agreement between researchers that have advanced 'rich' and 'lean' interpretations on past data on the interpretation of such successful performances. Our paradigm therefore offers one potential route towards mapping how planning abilities evolve.

Ethics. The methods of this study were carried out in accordance with the relevant guidelines and regulations. The study was conducted under approval from the University of Auckland Animal Ethics Committee (reference number 001823) and from the Province Sud with permission to work on Grande Terre, New Caledonia, and to capture and release crows.

Data accessibility. The full dataset is available on Open Science Framework OSF: https://osf.io/muaw9/?view_only=5908e4dcb90941be85a2d5344222ec9b.

Authors' contributions. Conceptualization: M.B., T.S., A.H.T. and N.S.C.; methodology: M.B., M.S., A.F., R.G., R.M., T.S., R.D.G., A.H.T. and N.S.C.; investigation: M.B., M.S., A.F. and R.G.; formal analysis: M.B.; writing, review and editing: M.B., M.S., A.F., R.G., R.M., T.S., R.D.G., A.H.T. and N.S.C.; supervision: A.H.T. and N.S.C.; funding acquisition: N.S.C. and A.H.T.; resources: R.D.G., N.S.C. and A.H.T.

Competing interests. All authors declare that they have no conflict of interest.

Funding. This research was supported by a Royal Society of New Zealand Rutherford Discovery Fellowship and a Prime Minister's McDiarmid Emerging Scientist Prize (A.H.T.), the European Research Council under the European Union's Seventh Framework Programme (FP7/2007-2013)/ERC Grant Agreement No. 3399933 (N.S.C.) and the Max Planck Institute for the Science of Human History (R.D.G.).

Acknowledgements. We thank Province Sud for granting us permission to work in New Caledonia and Dean M. and Boris C. for allowing us access to their properties for catching and releasing the crows.

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
