## [Reviewer comments · Proceedings of the Royal Society B: Biological Sciences]

Review History

RSPB-2020-1490.R0 (Original submission)

Review form: Reviewer 1 (John Marzluff)

Recommendation

Accept as is

Scientific importance: Is the manuscript an original and important contribution to its field?

Excellent

General interest: Is the paper of sufficient general interest?

Excellent

Quality of the paper: Is the overall quality of the paper suitable?

Good

Is the length of the paper justified?

Yes

Should the paper be seen by a specialist statistical reviewer?

No

Do you have any concerns about statistical analyses in this paper? If so, please specify them explicitly in your report.

No

It is a condition of publication that authors make their supporting data, code and materials available - either as supplementary material or hosted in an external repository. Please rate, if applicable, the supporting data on the following criteria.

Is it accessible?

Yes

Is it clear?

Yes

Is it adequate?

Yes

Do you have any ethical concerns with this paper?

No

Comments to the Author

Once again the ingenuity of a corvid and their investigators is demonstrated in a finely crafted, executed and interpreted experiment. While appearing quite simple, this experiment is subtly complex and accounts for prior issues with the spoon test. The birds significantly selected the right tool to solve a novel problem. In so doing they first had to understand the problem and solution, then remember that solution, and select a tool that would result in a high quality reward at a later time. Insight (problem solution) and especially foresight (memory from training that the task would be available in the future and that the correct tool must be selected beforehand to earn the reward) are both demonstrated. My only concern is the small sample size, which should be discussed more. I understand the logistics of such work and a sample of 6 birds is commendable. However, when all was done, only 3 of the 6 birds demonstrated an ability to learn the sequence or demonstrate foresight. The authors should discuss more fully why success of only half of the subjects is impressive. How does this compare with humans? Moreover, what are the characteristics of those that failed to learn the sequence and failed to demonstrate foresight? Were they predominantly of one sex? one age? or particular body condition? Motivation can play a large role in such experiments, so I wonder if those that did poorly were larger, of greater weight for their size, or shared other traits that may affect their willingness to participate or learn the experiment.

Review form: Reviewer 2

Recommendation

Major revision is needed (please make suggestions in comments)

Scientific importance: Is the manuscript an original and important contribution to its field?

Good

General interest: Is the paper of sufficient general interest?

Good

Quality of the paper: Is the overall quality of the paper suitable?

Acceptable

Is the length of the paper justified?

Yes

Should the paper be seen by a specialist statistical reviewer?

No

Do you have any concerns about statistical analyses in this paper? If so, please specify them explicitly in your report.

Yes

It is a condition of publication that authors make their supporting data, code and materials available - either as supplementary material or hosted in an external repository. Please rate, if applicable, the supporting data on the following criteria.

Is it accessible?

No

Is it clear?

N/A

Is it adequate?

N/A

Do you have any ethical concerns with this paper?

No

Comments to the Author

Please find my comments in the attached document. (See Appendix A)

Decision letter (RSPB-2020-1490.R0)

27-Jul-2020

Dear Dr Boeckle:

Your manuscript has now been peer reviewed and the reviews have been assessed by an Associate Editor. The reviewers' comments (not including confidential comments to the Editor) and the comments from the Associate Editor are included at the end of this email for your reference. As you will see, the reviewers and the Editors have raised some concerns with your manuscript and we would like to invite you to revise your manuscript to address them.

Research ethics:

Use of animals and field studies:

It is a condition of publication that you make available the data and research materials supporting the results in the article. Please see our Data Sharing Policies (<https://royalsociety.org/journals/authors/author-guidelines/#data>). Datasets should be deposited in an appropriate publicly available repository and details of the associated accession number, link or DOI to the datasets must be included in the Data Accessibility section of the article (<https://royalsociety.org/journals/ethics-policies/data-sharing-mining/>). Reference(s) to datasets should also be included in the reference list of the article with DOIs (where available).

Please submit a copy of your revised paper within three weeks. If we do not hear from you within this time your manuscript will be rejected. If you are unable to meet this deadline please let us know as soon as possible, as we may be able to grant a short extension.

Best wishes,
Dr Robert Barton
mailto: proceedingsb@royalsociety.org

Associate Editor

Comments to Author:

This is a really neat experimental design, and both referees praise the collaborative ethos of this paper. However, both require clarification of some key aspects of the methodology. Referee 2 also raises interesting points on the implications of trial design and training that should be addressed in the discussion.

Reviewer(s)' Comments to Author:

Referee: 1

Comments to the Author(s)

Once again the ingenuity of a corvid and their investigators is demonstrated in a finely crafted, executed and interpreted experiment. While appearing quite simple, this experiment is subtly complex and accounts for prior issues with the spoon test. The birds significantly selected the right tool to solve a novel problem. In so doing they first had to understand the problem and solution, then remember that solution, and select a tool that would result in a high quality reward at a later time. Insight (problem solution) and especially foresight (memory from training that the task would be available in the future and that the correct tool must be selected beforehand to earn the reward) are both demonstrated. My only concern is the small sample size, which should be discussed more. I understand the logistics of such work and a sample of 6 birds is commendable. However, when all was done, only 3 of the 6 bird demonstrated an ability to learn the sequence or demonstrate foresight. The authors should discuss more fully why success of only half of the subjects is impressive. How does this compare with humans? Moreover, what are the characteristics of those that failed to learn the sequence and failed to demonstrate foresight? Where they predominantly of one sex? one age? or particular body condition? Motivation can play a large role in such experiments, so I wonder if those that did poorly were larger, of greater weight for their size, or shared other traits that may affect their willingness to participate or learn the experiment.

Referee: 2

Comments to the Author(s)

Please find my comments in the attached document.

Author's Response to Decision Letter for (RSPB-2020-1490.R0)

See Appendix B.

RSPB-2020-1490.R1 (Revision)

Review form: Reviewer 2 (Zeynep Civelek)

Recommendation

Accept with minor revision (please list in comments)

Scientific importance: Is the manuscript an original and important contribution to its field?

Good

General interest: Is the paper of sufficient general interest?

Excellent

Quality of the paper: Is the overall quality of the paper suitable?

Excellent

Is the length of the paper justified?

Yes

Should the paper be seen by a specialist statistical reviewer?

No

Do you have any concerns about statistical analyses in this paper? If so, please specify them explicitly in your report.

No

It is a condition of publication that authors make their supporting data, code and materials available - either as supplementary material or hosted in an external repository. Please rate, if applicable, the supporting data on the following criteria.

Is it accessible?

Yes

Is it clear?

Yes

Is it adequate?

Yes

Do you have any ethical concerns with this paper?

No

Comments to the Author

Thank you very much for your detailed response to every point. I am sorry about the mix-up with regards to the follow-up, which happens! Overall, the methods are now all clear and the results are discussed more thoroughly.

One minor point with regards to the Subjects section is that in your response to the reviewer you explained the sample size/dropouts as: "Nine individuals entered training. Two individuals (Jupiter, Io) did not reach criterion in the tool functionality training and were therefore excluded from later procedures and testing. Four of the six crows tested were able to make correct choices in Conditions 1 and 2, with subjects taking on average of 22 trials to learn this. Two individuals (Mars, Venus) entered the training stage (C1&2) but did not reach criterion and were excluded from testing. One bird (Mercury) was excluded, due to experimenter error, it was moved onto the test phase (C3&4) before reaching the correct criterion in the training phase (C1&2). Four individuals reached criterion at C1 and C2 and then entered the testing phase (C3&4)." There seems to be a confusion in text- "Two individuals ('Mars', 'Mercury', 'Venus') entered the training stage (C1&2). Four individuals reached training criterion at C1 and C2 and then entered the testing phase (C3&4)." - that needs a minor edit.

Decision letter (RSPB-2020-1490.R1)

06-Oct-2020

Dear Dr Boeckle

I am pleased to inform you that your manuscript RSPB-2020-1490.R1 entitled "New Caledonian crows plan for specific future tool use" has been accepted for publication in Proceedings B.

The referee(s) have recommended publication, but also suggest some minor revisions to your manuscript. Therefore, I invite you to respond to the referee(s)' comments and revise your manuscript. Because the schedule for publication is very tight, it is a condition of publication that you submit the revised version of your manuscript within 7 days. If you do not think you will be able to meet this date please let us know.

- 1) A text file of the manuscript (doc, txt, rtf or tex), including the references, tables (including captions) and figure captions. Please remove any tracked changes from the text before submission. PDF files are not an accepted format for the "Main Document".
- 2) A separate electronic file of each figure (tiff, EPS or print-quality PDF preferred). The format should be produced directly from original creation package, or original software format. PowerPoint files are not accepted.
- 3) Electronic supplementary material: this should be contained in a separate file and where possible, all ESM should be combined into a single file. All supplementary materials

accompanying an accepted article will be treated as in their final form. They will be published alongside the paper on the journal website and posted on the online figshare repository. Files on figshare will be made available approximately one week before the accompanying article so that the supplementary material can be attributed a unique DOI.

If you wish to submit your data to Dryad (<http://datadryad.org/>) and have not already done so you can submit your data via this link [http://datadryad.org/submit?journalID=RSPB&manu=\(Document not available\)](http://datadryad.org/submit?journalID=RSPB&manu=(Document not available)) which will take you to your unique entry in the Dryad repository. If you have already submitted your data to dryad you can make any necessary revisions to your dataset by following the above link. Please see <https://royalsociety.org/journals/ethics-policies/data-sharing-mining/> for more details.

Sincerely,
Dr Robert Barton
Editor, Proceedings B
<mailto:proceedingsb@royalsociety.org>

Associate Editor:

Board Member: 1

Comments to Author:

Thank you, the revised text is clear and conservative, and openly reflects on comments from both referees. One of the referees has kindly reviewed the resubmission and notes one clarification.

Reviewer(s)' Comments to Author:

Referee: 2

Comments to the Author(s)

Thank you very much for your detailed response to every point. I am sorry about the mix-up with regards to the follow-up, which happens! Overall, the methods are now all clear and the results are discussed more thoroughly.

One minor point with regards to the Subjects section is that in your response to the reviewer you explained the sample size/dropouts as: "Nine individuals entered training. Two individuals (Jupiter, Io) did not reach criterion in the tool functionality training and were therefore excluded from later procedures and testing. Four of the six crows tested were able to make correct choices in Conditions 1 and 2, with subjects taking on average of 22 trials to learn this. Two individuals (Mars, Venus) entered the training stage (C1&2) but did not reach criterion and were excluded from testing. One bird (Mercury) was excluded, due to experimenter error, it was moved onto the test phase (C3&4) before reaching the correct criterion in the training phase (C1&2). Four individuals reached criterion at C1 and C2 and then entered the testing phase (C3&4)."

There seems to be a confusion in text- "Two individuals ('Mars', 'Mercury', 'Venus') entered the training stage (C1&2). Four individuals reached training criterion at C1 and C2 and then entered the testing phase (C3&4)." - that needs a minor edit.

Author's Response to Decision Letter for (RSPB-2020-1490.R1)

See Appendix C.

Decision letter (RSPB-2020-1490.R2)

12-Oct-2020

Dear Dr Boeckle

I am pleased to inform you that your manuscript entitled "New Caledonian crows plan for specific future tool use" has been accepted for publication in Proceedings B.

Your article has been estimated as being 8 pages long. Our Production Office will be able to confirm the exact length at proof stage.

Open Access

Paper charges

Sincerely,

Appendix A

Summary:

This study aims to examine whether New Caledonian crows select the correct tool among five options (a functional tool, two previously successful tools, a distractor and a low value food) to retrieve a high value reward from an apparatus that is going to be available in the future. Critically, different from the previous item choice tasks, all tools (except for the distractor) had previously been associated with the reward before. Therefore, a previously functional tool acted as a distractor in some trials and it was argued that the successful performance cannot be explained by associating a particular tool with the reward. The results showed that the subjects made the correct tool choice for an apparatus they observed earlier and that would be available to them in the future.

Strengths:

- This is an interesting topic that has created a lot of debate. It is great to see researchers from both sides of the debate working towards a better understanding of nonhuman animals' capabilities with regards to future planning.
- The background section is well written. It states the source of disagreements on the interpretation of success in the spoon task using specific examples. It is brief but comprehensive.
- The researchers suggest a novel way of approaching the question of whether nonhuman animals plan for specific future situations. The study is preregistered and the researchers seem to have adhered to the plans except for one instance I can identify (mentioned in the 'Recommendations for improvement' part).
- The methods are explained clearly and the figures are very professional and helpful in clarifying each condition.
- The results and analyses are clearly explained. The trial by trial data gives a good picture of the overall performance.

Recommendations for improvement:

Please clarify the sample size throughout. It is claimed that there are 9 crows to begin with and I suppose 3 of them did not pass the prior experience training but there is no information about this in the manuscript. (e.g., Line 177- "Each of the four crows..")

My main concern about this study is with regards to the prior experience training that were explained in detail in the Electronic Supplementary Materials (ESM).

1. While I understand there is not enough space for details in the manuscript, it is necessary to explicitly refer the readers to the ESM so that they get a fuller understanding of the crows' prior experiences other than those mentioned in the manuscript.
2. Can you please clarify what would be the minimum training required for this task? I believe the crows had to learn 1) the tool-apparatus pairs (tool functionality training in ESM), 2) the temporal order of events (Stages 1, 2 and 3) and to choose a tool and carry it from one compartment to another comfortably (tool selection and tool transport training in ESM) to be able to participate in the test phase of this study. However, in addition to these, they received five other training (i.e., a quality allocation training, apparatus functionality training, mental representation training, five choice tool functionality training, delay of gratification training and they participated in a meta-tool experiment).
3. With regards to the tool transport training, it was not clear in the ESM whether the crows needed to transport the correct tool from one compartment to the other which had an apparatus or could it be any tool? What was the criterion to pass this training?
4. In the 'quality allocation training', they were trained not to choose the low value apple over the meat even when it was readily available and meat needed to be extracted from an apparatus

with a tool. In the 'delay of gratification experience', this preference was further reinforced. Given these experiences to ignore the apple, I do not see how the apple may act as a distractor in Test phase as it was intended. The fact that none of the crows ever chose this reward in the reported trial by trial data is not surprising. Do you think a more stringent criteria than .20 should be used for the chance level analyses? In relation to this, can you clarify if/when the distractor item (a ball?) was presented to the crows prior to the Test phase? Trial by trial data shows an 'r' being chosen only 3 times by one of the crows but it does not mention what it refers to. I assume this is the distractor item and I am wondering if you found it unusual that it was chosen only very rarely and by one crow. Is it possible that they were neophobic and this item was never an option for the crows anyway?

5. In the other trainings the tube and platform apparatuses were used and the crows gained extensive experience to select/use the correct tool with the correct apparatus even though they had previously learnt the functional tools for these apparatuses (i.e., reaching the criterion in tool use training). Isn't it possible that the crows recognize and select a tool based on their extensive past exposure to certain tool-apparatus pairings that led to food and without imagining a future event (Stage 3)? I believe you aimed to test this in the follow-up experiment where the apparatus is taken away after 30 secs. However, differently to the preregistration form, you baited the apparatus with an apple instead of meat - the reward that they were trained to ignore and never even chose once when it was freely available. Unsurprisingly, the crows chose the freely available meat over a tool. Can you discuss this further in the paper?

Given the points raised above, the discussion section could benefit from speculating about the possible role of prior learning/associative learning in the crows' successful performance (as was shown in the Training phase). It may not be that "their performance was clearly not based on a preference for a specific tool type but on their observation of which problem they would have available to them in the future."

Minor points:

- Can you please provide the data for the prior experience training (ESM)? It could be in table form to show how long it took each subject to pass these phases., who dropped out/when.
- Can you please explain why the meta-tool experiment was reported in the ESM (i.e., in order to give the history of the crows' experimental history)?
- Line 61 main text: References (12 – 14) do not match with the statement and I believe these studies are not mentioned anywhere else in the review.
- Was it ever the case that the crows did not spontaneously take a tool with them to Compartment 1 (for Stage 3) even after selecting it or did not interact with the apparatus even when they had the tool? What was the procedure in these situations?
- In trials where the crows chose the meat (Condition 2), did they attempt to go back to the first compartment (e.g., was the door opened)? Or did the trial end there?
- Line 147: This paragraph seems like a repetition other than the last sentence.
- Line 157: There is no Table S3. Can you please add this?
- Line 20 in the ESM: "were trained"?
- Line 168: In line with the issue raised in the above section, I think this statement is a little misleading: "We chose apple as low-value immediate reward based on a preference test conducted in a previous study, in which they chose apple over tools." They were trained extensively not to choose the apple. Their preference was not spontaneous.
- What is your GLMM model for testing the learning effect? Did you control for trial type (apparatus type)? It seems to be a relevant variable as there is a lot of training involved with the platform and tube apparatuses but not so much with the dispenser.

- Can you provide a link for the movie in the ESM please?
- Why do you think the performance of the animals in the pilot study was worse than the current study? Can you refer to this pilot in your discussion of the findings?
- Figures 1 and 2 appear twice in the main text in different sizes.
- Please state what “r” represents in Table 3.

Appendix B

Reviewer(s)' Comments to Author:

Referee: 1

Comments to the Author(s)

Once again the ingenuity of a corvid and their investigators is demonstrated in a finely crafted, executed and interpreted experiment. While appearing quite simple, this experiment is subtly complex and accounts for prior issues with the spoon test. The birds significantly selected the right tool to solve a novel problem. In so doing they first had to understand the problem and solution, then remember that solution, and select a tool that would result in a high quality reward at a later time. Insight (problem solution) and especially foresight (memory from training that the task would be available in the future and that the correct tool must be selected beforehand to earn the reward) are both demonstrated. My only concern is the small sample size, which should be discussed more. I understand the logistics of such work and a sample of 6 birds is commendable. However, when all was done, only 3 of the 6 bird demonstrated an ability to learn the sequence or demonstrate foresight.

The authors should discuss more fully why success of only half of the subjects is impressive. How does this compare with humans?

Answer: We appreciate the concern about the small sample size and now explicitly discuss this: "A potential limitation of the study was the low number of individuals that were tested. Due to the space and time restrictions of the field season, it was not possible to include more than 9 individuals in this study, and only four of nine individuals passed the initial training criterion."

Note that our aim is to examine if crows can pass such tasks, not if all or if a certain proportion do so. As such, we do not think this is problematic (and there are many examples of studies using similarly small N when pursuing existence proof) as long as the findings are interpreted with due caution. There is currently no directly comparable human data. But, for your information, we have tested 3-5 year old children using a 2-choice version of this task, and found that 3 year-olds did not select correctly significantly above chance, while 4- and 5-year olds did. However, out of the 87 tested children 41 (47%) passed both test trials, a proportion not dissimilar as we found in the crows.

Moreover, what are the characteristics of those that failed to learn the sequence and failed to demonstrate foresight? Were they predominantly of one sex? one age? or particular body condition? Motivation can play a large role in such experiments, so I wonder if those that did poorly were larger, of greater weight for their size, or shared other traits that may affect their willingness to participate or learn the experiment.

Answer: Thanks a lot for the helpful feedback. We now note: "Interestingly, all individuals that succeeded in the test Conditions 3&4 were female, of which, only the individual 'Uranus' was categorized as an adult"

However, we refrain from discussing this further given the small number of birds tested. It is not possible to draw conclusions about individual differences.

Referee 2:

Summary:

This study aims to examine whether New Caledonian crows select the correct tool among five options (a functional tool, two previously successful tools, a distractor and a low value food) to retrieve a high value reward from an apparatus that is going to be available in the future. Critically, different from the previous item choice tasks, all tools (except for the distractor) had previously been associated with the reward before. Therefore, a previously functional tool acted as a distractor in some trials and it was argued that the successful performance cannot be explained by associating a particular tool with the reward. The results showed that the subjects made the correct tool choice for an apparatus they observed earlier and that would be available to them in the future.

Strengths:

- This is an interesting topic that has created a lot of debate. It is great to see researchers from both sides of the debate working towards a better understanding of nonhuman animals' capabilities with regards to future planning.
- The background section is well written. It states the source of disagreements on the interpretation of success in the spoon task using specific examples. It is brief but comprehensive.
- The researchers suggest a novel way of approaching the question of whether nonhuman animals plan for specific future situations. The study is preregistered and the researchers seem to have adhered to the plans except for one instance I can identify (mentioned in the 'Recommendations for improvement' part).
- The methods are explained clearly and the figures are very professional and helpful in clarifying each condition.
- The results and analyses are clearly explained. The trial by trial data gives a good picture of the overall performance.

Recommendations for improvement:

Please clarify the sample size throughout. It is claimed that there are 9 crows to begin with and I suppose 3 of them did not pass the prior experience training but there is no information about this in the manuscript. (e.g., Line 177- "Each of the four crows..")

Answer: Thanks for pointing this out. We have now included a paragraph explaining the detailed participation of the birds: Line 103: Nine individuals entered training. Two individuals (Jupiter, Io) did not reach criterion in the tool functionality training and were therefore excluded from later procedures and testing. Four of the six crows tested were able to make correct choices in Conditions 1 and 2, with subjects taking on average of 22 trials to learn this. Two individuals (Mars, Venus) entered the training stage (C1&2) but did not reach criterion and were excluded from testing. One bird (Mercury) was excluded, due to experimenter error, it was moved onto the test phase (C3&4) before reaching the correct criterion in the training phase (C1&2). Four individuals reached criterion at C1 and C2 and then entered the testing phase (C3&4)

My main concern about this study is with regards to the prior experience training that were explained in detail in the Electronic Supplementary Materials (ESM).

1. While I understand there is not enough space for details in the manuscript, it is necessary to explicitly refer the readers to the ESM so that they get a fuller understanding of the crows' prior experiences other than those mentioned in the manuscript.

Answer: We agree that the reader can better understand the prior experience when we explicitly refer to the ESM. We added a sentence to refer to the prior experience in the ESM. Line 127: All birds participated in various experiments before the presented study, which partially represent separate studies. The training specifically required for the current study are tool use training, tool selection training, apparatus functionality training, hook training and tool transport training, five choice tool functionality training, and temporal sequence training representing Condition 1 & 2. For a detailed and complete description of the prior experience and specific training stages please see below and the Electronic Supplementary Materials (ESM).

2. Can you please clarify what would be the minimum training required for this task? I believe the crows had to learn 1) the tool-apparatus pairs (tool functionality training in ESM), 2) the temporal order of events (Stages 1, 2 and 3) and to choose a tool and carry it from one compartment to another comfortably (tool selection and tool transport training in ESM) to be able to participate in the test phase of this study. However, in addition to these, they received five other training (i.e., a quality allocation training, apparatus functionality training, mental representation training, five choice tool functionality training, delay of gratification training and they participated in a meta-tool experiment).

Answer: It is correct that the minimal experience would have been lower than the prior experience of the individuals actually was. This was due to the fact that we have limited time each field season to run experiments, and as the presented study was the most complex study in the field seasons it was run in, we tested birds in less complex studies prior to this study. We now also cite the already published studies. We called the other experiences training even though they represent full studies. We report the necessary training stages and in the main text.

Tool use training, Tool Selection Training, Apparatus Functionality training, Hook training and Tool transport training, Five choice tool functionality training, Temporal Sequence. We now state this in the main text as presented in the previous answer (see Line 127 of the main text).

3. With regards to the tool transport training, it was not clear in the ESM whether the crows needed to transport the correct tool from one compartment to the other which had an apparatus or could it be any tool? What was the criterion to pass this training?

Answer: Thanks for spotting the missing criterion. We have introduced the training criterion in Line 74 of the ESM.

4. In the 'quality allocation training', they were trained not to choose the low value apple over the meat even when it was readily available and meat needed to be extracted from an apparatus with a tool. In the 'delay of gratification experience', this preference was further reinforced. Given these experiences to ignore the apple, I do not see how the apple may act as a distractor in Test phase as it was intended. The fact that none of the crows ever chose this reward in the reported trial by trial data is not surprising. Do you think a more stringent criteria than .20 should be used for the chance level analyses? In relation to this, can you clarify if/when the distractor item (a ball?) was presented to the crows prior to the Test phase? Trial by trial data shows an 'r' being chosen only 3 times by one of the crows but it does not mention what it refers to. I assume this is the distractor item

and I am wondering if you found it unusual that it was chosen only very rarely and by one crow. Is it possible that they were neophobic and this item was never an option for the crows anyway?

Answer: We now address this point in the following paragraph: “These birds selected the correct tool even though the distractors had been solutions in other conditions or items that could have been more immediately rewarding (ie, a piece of apple or a ball). We note that the latter were not selected by the birds in Conditions 1-6, even though they had interacted with the ball during a familiarization phase and apple was a daily diet food item, raising the concern that they may have learned to ignore these items over the training trials. However, even when conservatively reanalyzing the results of Conditions 3 and 4 as if the crows had only been offered 3 options rather than 5 (and so changing the probability of choosing the correct object by chance from 20% to 33%), we obtain the same finding: three of the four crows performed significantly above chance. Thus, our results are robust to the possibility that the distractor objects did not work as intended. Still, future studies might want to use two different types of low-value food items; one for training and a different one for testing or run a control where the distractor is the optimal choice.”

We discuss potential issues now in Line 254ff of the discussion.

Thanks for spotting the missing information. We now describe the presentation of the distractor item a small ball in line 403. We presented the ball one day before the first choice training.

Again thanks for spotting the missing detail. We explain that r signifies a rejection to choose a tool, and that the trial was stopped thereafter. See line 235.

5. In the other trainings the tube and platform apparatuses were used and the crows gained extensive experience to select/use the correct tool with the correct apparatus even though they had previously learnt the functional tools for these apparatuses (i.e., reaching the criterion in tool use training). Isn't it possible that the crows recognize and select a tool based on their extensive past exposure to certain tool-apparatus pairings that led to food and without imagining a future event (Stage 3)? I believe you aimed to test this in the follow-up experiment where the apparatus is taken away after 30 secs. However, differently to the preregistration form, you baited the apparatus with an apple instead of meat - the reward that they were trained to ignore and never even chose once when it was freely available. Unsurprisingly, the crows chose the freely available meat over a tool. Can you discuss this further in the paper?

Answer: We have now added these sentences to the manuscript to discuss this matter more:

“Finally, we cannot completely rule out that crows chose the correct tool because of some kind of associative learning. We do not think this possibility is likely, however, because the birds were trained in C1 and C2 on a different apparatus combination than used during testing in C3 and C4, and we used temporal gaps in our study: the tools were presented 5 minutes after the presentation of the apparatus (when it was now out-of-sight) and crows were then only able to gain reward (if they had chosen the correct tool) 10 minutes after this. Furthermore, tools

acted as the functional choice in one trial but as distractor object in the next trial, so the birds could not succeed by simply selecting whatever tool was most recently associated with reward. To strengthen the case further, future studies could run control conditions where the apparatus is visibly removed or destroyed after Stage 1, to examine if the birds would continue to pick the now no longer functional tool, or indicate their understanding by switching to the lower-value apply option.”

We also note that there is a discrepancy between our pre-reg and our experiment in regard to our follow-up experiment. We had intended to use meat rather than apple for this condition, but due to a communication mix-up apple was used instead. This is why we now only mention this experiment in our SI, and why we do not discuss this finding in our main discussion.

We conducted the follow-up as specified in the pre-registration. Unfortunately, we did not explain the food presentation on the 5 choice stage in the preregistration. Additionally, we did not present the data for the follow-up in the main text as we agree that the results are ambiguous. Therefore we left the data in the ESM.

Given the points raised above, the discussion section could benefit from speculating about the possible role of prior learning/associative learning in the crows' successful performance (as was shown in the Training phase). It may not be that “their performance was clearly not based on a preference for a specific tool type but on their observation of which problem they would have available to them in the future.”

Answer: We now discuss this point in the paragraph above

Minor points:

- Can you please provide the data for the prior experience training (ESM)? It could be in table form to show how long it took each subject to pass these phases., who dropped out/when.

Answer: We published the data of most of the experiments and there are still more to be published. Therefore, we would like to publish the data in the respective published or upcoming publications. We indicate the training phases that were already published now in the ESM.

- Can you please explain why the meta-tool experiment was reported in the ESM (i.e., in order to give the history of the crows' experimental history)?

Answer: Similarly, to the answer above, most of the trainings were part of the training history. Published data referred to now with a citation.

- Line 61 main text: References (12 – 14) do not match with the statement and I believe these studies are not mentioned anywhere else in the review.

Answer: Thanks again for spotting this detail. We corrected the references now fitting to the statement.

- Was it ever the case that the crows did not spontaneously take a tool with them to Compartment 1 (for Stage 3) even after selecting it or did not interact with the apparatus even when they had the tool? What was the procedure in these situations?

Answer: The only bird that did not choose a tool was Saturn. Those trials are indicated with r in Table 3. These trials were terminated and counted as wrong choices. All other birds did select a tool in all trials. When the wrong tool was chosen, birds were able to interact with the chosen tool and the apparatus. We report this in Line 173ff.

- In trials where the crows chose the meat (Condition 2), did they attempt to go back to the first compartment (e.g., was the door opened)? Or did the trial end there?

Trials in C2 in which birds chose meat were terminated after the choice of meat. We now report this in Line 160.

- Line 147: This paragraph seems like a repetition other than the last sentence.

Answer: Thanks for spotting this detail. We deleted the repetition and left the last sentence.

- Line 157: There is no Table S3. Can you please add this?

Answer: Thanks again for spotting the detail. We used to have table S3 in the supplement but introduced it now in the main text. Trials in which we had to put the tool in a reachable position are indicated with an ' in table 3.

- Line 20 in the ESM: "were trained"?

Answer: Thanks for spotting the missing verb; we added trained in the sentence.

- Line 168: In line with the issue raised in the above section, I think this statement is a little misleading: "We chose apple as low-value immediate reward based on a preference test conducted in a previous study, in which they chose apple over tools." They were trained extensively not to choose the apple. Their preference was not spontaneous.

Answer: We agree and discuss this now in Line 254ff: "These birds selected the correct tool even though the distractors had been solutions in other conditions or items that could have been more immediately rewarding (i.e., a piece of apple or a ball). We note that the latter were not selected by the birds in Conditions 1-6, even though they had interacted with the ball during a familiarization phase and apple was a daily diet food item, raising the concern that they may have learned to ignore these items over the training trials. However, even when conservatively reanalyzing the results of Conditions 3 and 4 as if the crows had only been offered 3 options rather than 5 (and so changing the probability of choosing the correct object by chance from 20% to 33%), we obtain the same finding: three of the four crows performed significantly above chance. Thus, our results are robust to the possibility that the distractor objects did not work as intended. Still, future studies might want to use two different types of low-value food items; one for training and a different one for testing

or run a control where the distractor is the optimal choice.”

- What is your GLMM model for testing the learning effect? Did you control for trial type (apparatus type)? It seems to be a relevant variable as there is a lot of training involved with the platform and tube apparatuses but not so much with the dispenser.

Answer: We reran the models with apparatus type as random and fixed factor. Unfortunately, based on the small number of trials it was not possible to include the variable without overfitting or problems with the Hessian matrix.

- Can you provide a link for the movie in the ESM please?

Answer: The temporary link is:

<https://www.dropbox.com/s/9hsei8z3bb5ia1k/Neptune%2B4%2BTrials%2BC3%2526C4%2BSupplement%2BV10.mp4?dl=0>

Until we can publish the permanent link to the University of Cambridge repository.

- Why do you think the performance of the animals in the pilot study was worse than the current study? Can you refer to this pilot in your discussion of the findings?

Answer: We now refer to the previous study presented in the ESM and discuss the additional value of the current study. Additionally, we wanted to conduct a pre-registered study, which was not possible in the first year, where we had to develop the method.

- Figures 1 and 2 appear twice in the main text in different sizes.

Answer: The figures are only once in the main text. The second appearance is from the upload as files with the figure captions on the platform of the journal.

- Please state what “r” represents in Table 3.

Answer: Thanks for spotting this detail. We added that the r signifies a rejection to choose a tool.

Appendix C

Referee: 2

Comments to the Author(s)

Thank you very much for your detailed response to every point. I am sorry about the mix-up with regards to the follow-up, which happens! Overall, the methods are now all clear and the results are discussed more thoroughly.

One minor point with regards to the Subjects section is that in your response to the reviewer you explained the sample size/dropouts as: "Nine individuals entered training. Two individuals (Jupiter, Io) did not reach criterion in the tool functionality training and were therefore excluded from later procedures and testing. Four of the six crows tested were able to make correct choices in Conditions 1 and 2, with subjects taking on average of 22 trials to learn this. Two individuals (Mars, Venus) entered the training stage (C1&2) but did not reach criterion and were excluded from testing. One bird (Mercury) was excluded, due to experimenter error, it was moved onto the test phase (C3&4) before reaching the correct criterion in the training phase (C1&2). Four individuals reached criterion at C1 and C2 and then entered the testing phase (C3&4)."

There seems to be a confusion in text- "Two individuals ('Mars', 'Mercury', 'Venus') entered the training stage (C1&2). Four individuals reached training criterion at C1 and C2 and then entered the testing phase (C3&4)." - that needs a minor edit.

Thanks for spotting this detail! We changed it accordingly in line 104-110:

"Nine individuals entered training. Three individuals ('Jupiter', 'Io', 'Mercury') did not reach criterion in the tool functionality training and were therefore excluded from later procedures and testing. Two crows were unable to make correct choices in Conditions 1 and 2 ('Mars', 'Venus'). The remaining subjects took on average 22 trials to reach training criterion. Four individuals ('Saturn', 'Neptune', 'Triton', 'Uranus'), reached training criterion at C1 and C2 and then entered the testing phase (C3&4) and three individuals were tested in the follow-up ('Neptune', 'Triton', 'Uranus')."